



# Prediction of Crustal Dynamics for the Yishu Fault Zone Based on Slip Susceptibility Analysis

Guiyun Gao[1,2], Pu Wang[3], Chenghu Wang[1,2*], Chengwei Yang[1,2]

[1] Key Laboratory of Compound and Chained Natural Hazards Dynamics, National Institute of Natural Hazards, Ministry of
Emergency Management of China, Beijing 100085, China
[2] Beijing Engineering Research Center of Earthquake Observation, Beijing 100085, China.
[3] State Key Laboratory of Hydraulic Engineering Simulation and Safety, Tianjin University, Tianjin 300350, China

*Correspondence to*: Chenghu Wang (chenghuwang@ninhm.ac.cn)

**Abstract.** We estimate the slip susceptibility of faults in the Yishu fault zone using a slip tendency analysis based on coupled

tectonic stress profiles. Studying *in situ* stress data and focal mechanism solution data collected in this area over more than 30 years, we construct the stress profiles of the study area using the coupling analysis method for deep and shallow stress data. Subsequently, the slip susceptibility of the faults is comprehensively evaluated based on various stress indicators and other related influencing factors. Our results show that $\mu_m$ is low in the deep and shallow crust of the Yishu fault zone, indicating a relatively low degree of overall stress accumulation. A comprehensive evaluation of the susceptibility of fault slips based on

five typical influencing factors ($\mu_m$, $K$, $\theta$, S-wave velocity, and seismic density) indicates that the overall seismic risk in the central part of the Yishu fault zone is not high, while the northwestern Yishu fault zone exhibits high seismic risk. The southeastern part of the Yishu fault zone reflects the transition from medium to low seismic risk. These results provide geomechanical and fault mechanics evidence for evaluating the regional crustal dynamics of the Yishu fault zone.

## 1 Introduction

The Yishu fault zone, located in Shandong, is part of the Tanlu fault zone, which is the largest fault zone in eastern China (Figure 1). The fault zone begins in Tancheng, Shandong Province, in the south and extends to Laizhou Bay in the north, with an overall length of approximately 360 km and a strike of N10 °–25 °E (Fang et al., 1980). It consists of several main parallel faults, from which very complex branch fault structures are developed. Since the Quaternary, several main faults in the Yishu fault zone have experienced active faulting and strong seismic activity. Holocene activity in the eastern part of the fault zone

is also intensive, whereas late Quaternary activity in the western fault is not obvious. The Yishu fault zone is the source from which almost all powerful earthquakes, i.e., of magnitude 7 or greater, in the Tanlu fault zone, have been recorded. According to previous reports (Zhang et al., 2013b), an earthquake with magnitude M8.5 and seven earthquakes with magnitudes M7.0– 7.9 have occurred in the study area. The faulting styles of historical earthquakes in the Yishu fault zone have been dominated by strike-slip faulting, followed by thrust and normal faulting events (Wang et al., 2015b). The study area (Figure 1), which

occurs in the seismic gap area of the Yishu fault zone, has experienced few earthquakes in recent years (Zhang et al., 2013a).



Previous researches have directly or indirectly shown that stress accumulated in the region has not been released; as such, this is a dangerous part of the Yishu fault zone (Zhang et al., 2013a; Li et al., 2014; Zheng et al., 2015; Wang et al., 2020; Feng et al., 2017; Li et al., 2019).

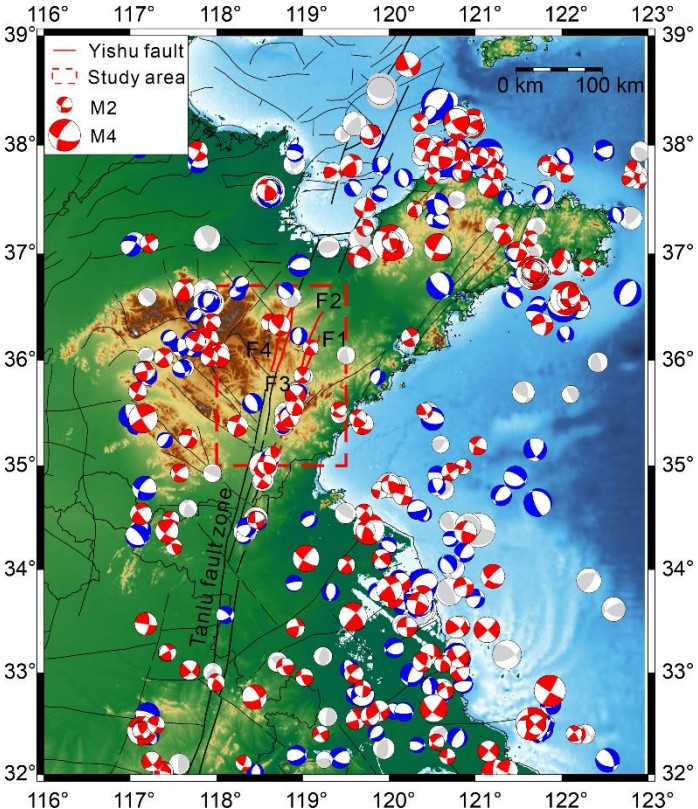

**Figure 1: Geological map of the Tanlu fault zone and the Yishu fault zone. The Yishu fault zone and the study area are indicated by the red line and the red dot line, respectively. The red, blue, and gray beach balls are the focal mechanisms of the strike-slip, normal, and thrust events, respectively. The Yishu fault zone consists of four main faults, which from east to west are the Changyi-Dadian fault (F1), the Anqiu-Juxian fault (F2), the Yishui-Tangtou fault (F3), and the Tangwu-Gegou fault (F4).**

Analysis of the mechanical process of seismogenic faults is important for understanding earthquake occurrence (Scholz, 2019). Many studies have addressed seismic risk in the Yishu fault zone, and scientists have repeatedly analyzed and studied the tectonic stress characteristics of the Yishu fault zone based on focal mechanism solutions (Xu et al., 1983; Zhou and Wei, 1987; Zhou et al., 2005; Dong et al., 1999; Cui and Xie, 2001; Zheng et al., 2013; Wang et al., 2015b; Yang et al., 2017). These results demonstrate that the Yishu fault zone presents a relatively uniform regional stress field in which the orientations of the maximum and minimum principal stresses are N82 E and N172 E, respectively. There is a regional transition of the orientation of the maximum horizontal principal compressive stress from the NEE to the near EW direction. This transition occurs smoothly and from west to east, such that the orientation of the maximum horizontal principal compressive stress gradually deflects from NEE to near EW. The stress regime is dominantly strike-slip faulting, but local differences have been noted.



In recent years, a large number of *in situ* hydraulic fracturing stress measurements in deep boreholes in the study area have

also shown that the direction of the regional stress field is NEE to nearly EW (Zhang et al., 2013a; Zheng et al., 2015; Feng et al., 2017; Li et al., 2019; Wang et al., 2020). Fault stability analysis based on *in situ* stress data shows that the Yishu fault zone is in a critical state with significant stress accumulation (Zhang et al., 2013a; Feng et al., 2017; Li et al., 2019; Liu et al., 2017); however, these measured data are relatively shallow compared to the depth of earthquake nucleation, and deep stress data are needed to provide more direct evidence. Zhao et al. (2016) used historical seismic records to compare and analyze seismic gap

images before the Tancheng M8.5 earthquake in 1668 and in modern East China, showing similarity between them, indirectly showing that the Tanlu fault zone is currently in the stress accumulation stage (Zhao et al., 2016). By comparing and analyzing measured *in situ* stress data for shallow area and focal mechanism solutions in deep area (>1km), it can be seen that deep and shallow crusts within the Yishu fault zone are affected by a uniform and stable stress field and that no stress field decoupling phenomenon is present (Zang and Stephansson, 2010). To combine the measured data of regional *in situ* stress with the focal

mechanism solution, Wang et al. (2019) proposed the stress polygon method using the horizontal-to-vertical in situ stress ratio (defined by $K_{max} = S_{Hmax}/S_V$ and $K_{min} = S_{hmin}/S_V$) combined with the stress shape ratio $R (= (S_1 - S_2)/(S_1 - S_3))$ from the focal mechanism solutions to obtain detailed constraints on the magnitudes of stress in deep depth. Therefore, the ranges of the maximum and minimum horizontal principal stress values were obtained at three depths in the Zijingguan area, Yixian County, Hebei Province (Wang et al., 2019).

The analysis of fault slip susceptibility is developed from crustal stability analysis based on geomechanical theory. Sibson (1985) introduced the effective stress ratio of the maximum and minimum principal stresses ($=\sigma_1/\sigma_3$) as a means of analyzing fault slip susceptibility (Sibson, 1985); however, this is a 2D method requiring that the azimuth of the intermediate principal stress must be parallel to the fault plane. Morris et al. (1996) proposed another method for the characterization of fault slip susceptibility, whereby the slip susceptibility of a fault is characterized by slip tendency $T_s$, which is defined as the ratio of

shear stress to effective normal stress ($T_s = \tau/(\sigma_n - p_0)$) (Morris et al., 1996). The ratio of maximum shear stress to the mean principal stress (defined by $\mu_m = ((S_1-S_3)/2)/((S_1 + S_3)/2 - p_0)$) was also introduced to evaluate the strength and level of stress accumulation of a fault and thus determine its slip susceptibility (Wang et al., 2014). Zhang et al. (2013a) calculated the upper and lower limits of the critical value of the maximum horizontal principal stress during fault activity using the Coulomb criterion based on measured *in situ* stress data. They showed that the stress value measured near the midsection of the Tanlu

fault zone did not reach its critical value, indicating that this area was in a stable state with low earthquake possibility. Feng et al. (2017) analyzed the results of the hydraulic fracturing tests of two boreholes in Shandong province, concluding that the stress accumulation level is the highest in the southern part of the Yishu fault zone, which may be close to the critical state of the fault slip.

The slip susceptibility characteristics of faults are closely related to earthquake occurrence, but few studies have systematically

analyzed the slip susceptibility characteristics of regional faults as a means to evaluate the risk of earthquakes from coupled tectonic stress profiles. Given the importance of the Yishu fault zone for crustal dynamics research and earthquake prevention and disaster mitigation in North China, here we predict its crustal dynamics based on slip tendency analysis based on coupled



tectonic stress profiles. Using *in situ* stress data and focal mechanism solution data collected in this area for more than 30 years, we construct the stress profile of this area by coupling deep and shallow stress data as we initially proposed (Wang et al.,

2019). Subsequently, the constructed stress profiles are used to analyze the fault slip susceptibility in the Yishu fault zone, providing geomechanical and fault mechanics evidence that enables the evaluation of the regional trend of crustal dynamics within the Yishu fault zone.

## 2 Theory of fault stability

The analyses of fault slip susceptibility are based on the classical Mohr-Coulomb stress criterion (Zoback, 2007; Chen, 1988),

Anderson's theory (Anderson, 1951) and Byerlee's law (Byerlee, 1978). Using a large number of experimental data, Coulomb proposed in 1785 that, in addition to the friction coefficient $\mu$, cohesion $C_0$ should also be taken into account to describe the mechanical process of rock sliding friction.

$$\tau = C_0 + \mu \sigma_n \tag{1}$$

where $\tau$ is shear stress and $\sigma_n$ is normal stress. On the basis of this equation, a large volume of research has been conducted to

understand the friction strength and behavior of rocks. The strength of the crust or a fault is often described by its friction coefficient (Muluneh et al., 2018). Byerlee concluded that the friction coefficient of the fault was between 0.6 and 0.85 after summarizing a large number of experimental data on different types of rocks and faults, thus developing Byerlee's law. The influence of cohesion $C_0$ is often ignored so that a simplified formula can be used to calculate the apparent friction strength $\mu$ of rocks or faults as follows:

$$\mu = \tau / \sigma_n \text{ or } \mu = \tau / (\sigma_n - p_0) \tag{2}$$

where $P_0$ is the pore pressure. The apparent friction strength $\mu$ of the faults is an important indicator that can be used to determine the strength of a fault when the stress state is known. It can be approximated to the friction strength of the fault where there is relative sliding between the two sides of the fault. The apparent friction strength $\mu$ also called slip tendency by Morris et al. (1996) and proposed as an indicator for the fault slip susceptibility.

For natural faults, the relationship between the orientation of the fault and the three principal stresses that intersected the region in which it is located is reflected in the projection expression method of the principal stress tensor on the fault plane. As shown in Figure 2, the $S_1$, $S_2$, and $S_3$ axes of the applied coordinate system represent the orientations of the three principal stress axes. When the regional *in situ* stress state is known, the normal and shear stresses on a specific fault plane can be calculated using the projection relationship of the stress tensor on the fault plane. The detailed calculation formula (Jaeger et al., 2009) is as

follows:

$$\sigma_n = l^2 S_1 + m^2 S_2 + n^2 S_3$$
$$\tau = \sqrt{(l S_1)^2 + (m S_2)^2 + (n S_3)^2 - \sigma_n^2} \tag{3}$$



where $S_1$, $S_2$, and $S_3$ are the maximum, intermediate, and minimum principal stresses, respectively; and $l = \cos\phi$, $m = \cos\gamma$, and $n = \cos\theta$ are the directional cosines of the three principal stress axes. With regard to the effect of the pore fluid pressure, the effective normal stress is $\sigma'_n = \sigma_n - P_0$. The additional hydrostatic stress gives no rise to shear stress on any plane, so the

effective shear stresses are identical to the actual shear stresses.

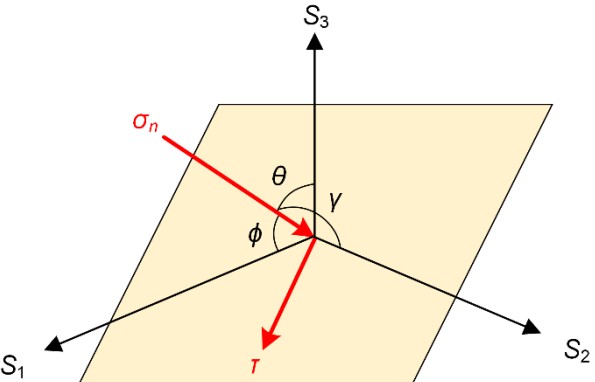

**Figure 2: Schematic diagram of the coordinate system used for the calculation of shear stress $\tau$ and normal stress $\sigma_n$ on the fault plane. $S_1$, $S_2$ and $S_3$ are the maximum, intermediate and minimum principal stresses, respectively. $\phi$, $\gamma$, and $\theta$ are the angle between the normal direction of the fault plane and the three principal stress axes, respectively.**

Anderson (1905) assumed that the three principal stress directions in the crust are either vertical or horizontal, then divided the faults into three basic faulting styles according to relationships between the geometrical morphology of a fault and the stresses that create the fault.

$$S_v > S_H > S_h, \quad \text{Normal faulting } (NF) \text{ stress regime}$$
$$S_H > S_h > S_v, \quad \text{Reverse faulting } (RF) \text{ stress regime} \tag{4}$$
$$S_H > S_v > S_h, \quad \text{Strike-slip } (SS) \text{ faulting stress regime}$$

where $S_H$ is the maximum horizontal principal stress, $S_h$ is the minimum horizontal principal stress, and $S_V$ is the vertical stress.

Anderson's theory provides a simplified tectonic stress field near the fault after ignoring the intermediate principal stress, which is, therefore, essentially a two-dimensional *in situ* stress field. Depending upon the type of faulting considered, the expressions of normal and shear stresses on the fault plane can be further simplified. Combined with Eq. 4, the friction strength can be obtained, as shown in Eq. 5, in which $S_1$, $S_2$, and $S_3$ correspond to $S_H$, $S_h$, and $S_V$ according to the faulting type.

$$\mu = \begin{cases} \dfrac{\sqrt{l^2 S_v^2 + m^2 S_H^2 + n^2 S_h^2 - \left(l^2 S_v + m^2 S_H + n^2 S_h\right)^2}}{l^2 S_v + m^2 S_H + n^2 S_h - P_0} & (NF) \\[4mm] \dfrac{\sqrt{l^2 S_H^2 + m^2 S_v^2 + n^2 S_h^2 - \left(l^2 S_H + m^2 S_v + n^2 S_h\right)^2}}{l^2 S_H + m^2 S_v + n^2 S_h - P_0} & (SS) \\[4mm] \dfrac{\sqrt{l^2 S_H^2 + m^2 S_h^2 + n^2 S_v^2 - \left(l^2 S_H + m^2 S_h + n^2 S_v\right)^2}}{l^2 S_H + m^2 S_h + n^2 S_v - P_0} & (RF) \end{cases} \tag{5}$$



Crustal rupture or fault slip phenomena are essentially examples of shear failure, which means that the slip failure is primarily
caused by the continuous accumulation of shear stress in the crust. Townend and Zoback (2000) found that the crust maintains
a limited state of relatively balanced rupture through fault slip or seismic activity (Townend and Zoback, 2000). The ratio of
the maximum shear stress ($(S_1-S_3)/2$) to the effective mean stress ($(S_1 + S_3) / 2 - P_0$) on a certain fault plane, that is, the frictional
characteristic index $\mu_m$, is used to assess the degree of stress accumulation or crustal stability. The larger the value of $\mu_m$, the

higher the degree of stress accumulation (Wang et al., 2014).

If the fault is in accordance with Byerlee-Anderson theory, the relationship between $\mu_m$ and the apparent friction coefficient of
the fault $\mu$ can be given as:

$$\mu_m = \frac{(S_1 - S_3)/2}{(S_1 + S_3)/2 - P_0} = \frac{\mu}{\sqrt{1 + \mu^2}} \; . \tag{6}$$

The crustal stresses are maintained in a critical equilibrium state through fault movement and seismic activity. Differential

stress and mean stress remain in an almost constant ratio under the critical stressed state. Thus, the $\mu_m$ can reflect the frictional
strength of a fault from the viewpoint of stress accumulation (Wang et al., 2014; Wang et al., 2015a). According to the
Anderson fault theory system, there is a positive correlation between the apparent friction strength of the fault $\mu$ and $\mu_m$ (Eq.
6). When the apparent friction strength of the fault is less than 0.6, the magnitudes of the two are very close (Wang et al.,
2015a). Therefore, it is convenient to calculate the $\mu_m$ value and use it to estimate the magnitude of the friction strength of the

fault when abundant measured stress data are available and the spatial orientation of the fault is unknown.

The regional fault friction strength can also be estimated based on focal mechanism solutions. Iio (1997) proposed a method
for the macroscopic analysis of regional fault friction strength using focal mechanism solutions. The seismic fault slip is
considered to occur only on the dominant slip plane and on the fault with the largest Coulomb stress in the direction of the
fault slip. At maximum Coulomb stress, the relationship between the fault friction strength $\mu$ and the angle $\theta$ between the

principal stress and the fault plane is as follows (Iio, 1997):

$$\theta = \frac{1}{2} \arctan \frac{1}{\mu} \tag{7}$$

Therefore, the direction of the maximum principal stress of the fault plane can be determined by the friction strength of the
fault. For example, when the friction coefficients are 0, 0.3, and 0.6, $\theta$ are 45 °, 37 °, and 30 °, respectively. The angle between
the $P$ axis of the focal mechanism solution and the fault plane is always 45 °; thus, the friction strength of the fault is relatively

large when the $P$-axis direction is inconsistent with the direction of the maximum principal stress. Figure 3 shows a schematic
diagram of the relationship between the $P$-axis distribution and the fault friction strength in two special cases.



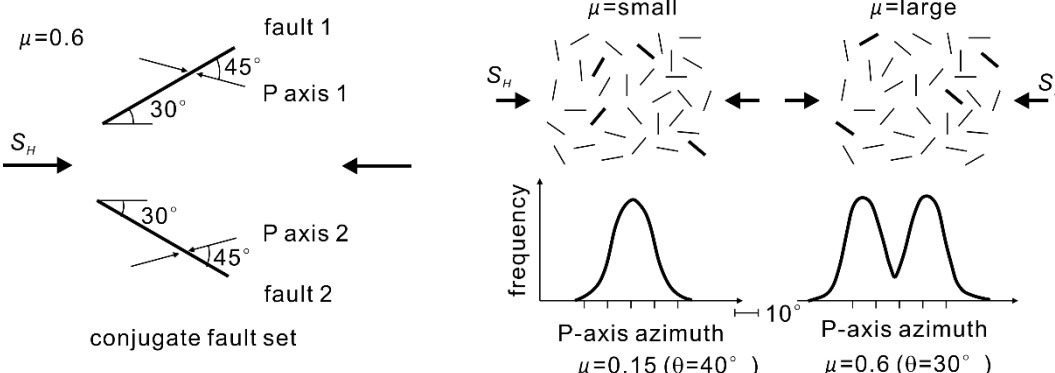

**Figure 3: Relationship between the *P* axis distribution and the fault friction strength revised from (Iio, 1997). The relationship between the azimuth of the *P* axis and the direction of the maximum horizontal principal stress $S_H$ is illustrated on the left when $\mu$ is 0.6. The maximum horizontal principal stress $S_H$ is assumed to work on a vertical strike-slip fault plane having various azimuths. At two different mean values of $\mu$, i.e. $\mu$=0.15 and $\mu$=0.6 the distributions of the *P* axes are quite different. The standard deviation of the *P*-axis azimuth is 10 °and only favorably oriented faults, marked in bold lines, fail. The distribution of the *P* axes shows a narrow peek for the smaller value of $\mu$, while two peeks are shown for the larger value of $\mu$.**

The orientation of the *P* axis in Figure 3 refers to the angle between the *P* axis and the direction of the maximum horizontal principal stress. The value of the fault friction strength $\mu$ is not fixed, but distributed around a certain value. The orientation of the *P* axis shows a narrow peek distribution centered on the direction of the maximum horizontal principal stress when $\mu$ is small. Therefore, the orientation of the *P* axis will thus be far away from the direction of the maximum horizontal principal stress (±15 °) in a 'bimodal' distribution with two peaks when $\mu$ is large. Based on this analysis, the regional fault friction strength can be determined macroscopically.

## 3 Regional geological background and fault structure

The Yishu fault zone is located in the middle part of the Tanlu fault zone. Four main faults with strikes of NNE run through the study area; From east to west, these are the Changyi-Dadian fault (F1), the Anqiu-Juxian fault (F2), the Yishui-Tangtou fault (F3), and the Tangwu-Gegou fault (F4). The Anqiu-Juxian graben lies between F1 and F2. The Gongdan mountain horst is between F2 and F3 fault, and the Mazhan-Sucun graben lies between F3 and F4 (Chao et al., 1998). The spatial distribution characteristics of the faults are shown in Figure 1.

The slip susceptibility analysis of regional faults requires relatively precise knowledge of their spatial distribution characteristics, whereupon the potential slip possibility of the studied faults can be quantified using the fault mechanics theory introduced above. In this study, we quantitatively determined the spatial orientation characteristics of regional faults from shallow to deep depths using data obtained by three methods. For outcrops of faults exposed near the surface, the orientation of the main faults can be obtained by ground survey and trench exploration. The orientations of the main faults in the Yishu fault zone were collated from the literature (Wang et al., 2011) and are listed in Table 1. We then quantified the dominant strike angle and the dip angle of the regional geological structure from the shallow surface to depths of 400 m using high-precision ultrasonic borehole television. We also used recent results for Yishu fault zones, including traditional geological





profiles (Fu et al., 2017; Geng et al., 2019), seismic reflection profiles (Liu et al., 2015), and electrical profiles (Hu, 2014), and comprehensive geophysical profiles (Xu et al., 2022). To determine the deep structure of the fault, seismic deep reflection profiles provide the most effective method of detection. We collected and evaluated information on the deep seismic reflection profile of the faults in the study area in recent years (Qin et al., 2020; Xu et al., 2022; Duan et al., 2015) using data obtained by multiple survey lines through the Yishu fault zone to clearly define the spatial orientation of the deep fault structure. The evaluated results show that the deep structure of the Yishu fault is a strike-slip fault with a high dip angle of $70°-90°$ and a

strike of $0°-30°$ (Figure 4). Seismic information can also enable the structure of active faults to be determined. By analyzing the comprehensive solution information of the focal mechanism at different depths in the study area, the information about the occurrence close to the true cross-sectional solution of the focal mechanism is found to be consistent with the real fault information obtained from the deep reflection profile. The fault strikes are concentrated mainly between $10°$ and $30°$, with an average of $17°$. And the dip angles are mainly concentrated between $70°$ and $80°$, with an average of $73°$. Therefore, the

complete information about the occurrence of deep faults can be determined.

The shallow outcrop of the fault zone in the study area, the structural plane information of the central borehole, the deep reflection profile, and the source information were unified and analyzed to construct a complete fault occurrence profile (Figure 4). As shown in Figure 4(b), the fault dip angles measured by different methods are concentrated between $60°$ and $80°$, with an average dip angle of $72°$. The strikes of faults identified in the shallow outcrop and deep reflection profile are consistent

with each other, ranging from $0°$ to $30°$. Since the structural plane interpreted by the borehole observation contains hidden faults with abrupt strikes, the fault strike increases in the range of 100 to 1000 m, mainly distributed from $40°$ to $90°$. As shown in Figure 4, although abrupt changes occur at some depths, the overall fault occurrence does not fluctuate significantly with depth. The average occurrence of the faults can be preliminarily determined as $12°$ in the strike angle and $72°$ in the dip angle.

**Table 1: Summary of the fault information in the Yishu fault zone (Wang et al., 2011)**

| No. | Fault name | Length /km | Attitude | Faulting type | Time of recent activity |
|---|---|---|---|---|---|
| 1 | Zhangdian-Renhe fault | 50 | 335°/225°∠70°−85° | left-normal-slip fault | Q_{3-2} |
| 2 | Wangmushan fault | 60 | 25°∠W | right-normal-slip fault | Q_2 |
| 3 | Yuwangshan fault | 90 | SN/E∠60°−80° | right-normal-slip fault | Q_2 |
| 4 | Guangrao-Qihe fault | 175 | NEE/NW∠20°−70° | normal fault | Q_{1-2} |
| 5 | Zihe fault | 90 | NNE/E∠60°−80° | right-reverse-slip fault | Q_{1-2} |
| 6 | Yidu fault | 140 | 330°/80°∠70° | left-normal-slip fault | Q_3 |
| 7 | Shuangshan-Lijiazhuang fault | 190 | 340°/W∠65°−80° | right-normal-slip fault | Q_{3-4} |
| 8 | Shangwujing fault | 130 | 40°/NW∠60°−80° | right-normal-slip fault or right-reverse-slip fault | Q_3 |
| 9 | Changyi-Dadian fault | 150 | 10°−20°/NWW∠60°−80° | right-normal-slip fault or right-reverse-slip fault | Q_3 |
| 10 | Anqiu-Juxian fault | 150 | 10°−20°/SEE∠40°−80° | right-normal-slip fault or right-reverse-slip fault | Q_{3-4} |



| 11 | Yishui-Tangtou fault | 150 | 10°/NWW∠60°−80° | right-normal-slip fault | Q$_{2-3}$ |
| 12 | Tangwu-Gegou fault | 150 | 5°−15°/SEE∠70° | right-normal-slip fault | Q$_{2-3}$ |


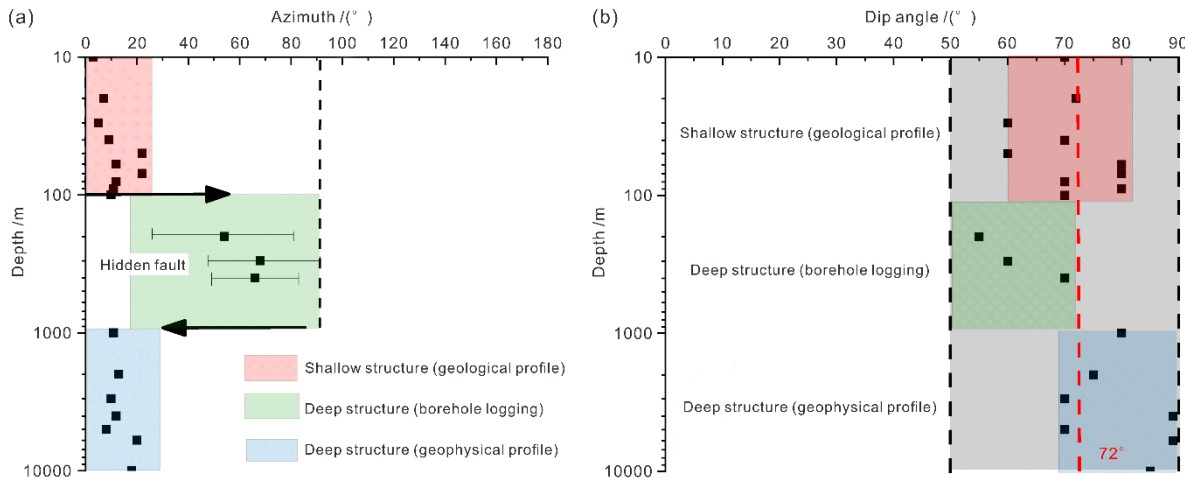

**Figure 4: Summary of fault occurrence information in the Yishu fault zone. (a) Comprehensive fault strike statistics for shallow and deep structures. (b) Comprehensive fault dip angle statistics for shallow and deep structures.**

## 4 Multi-source stress data and regional stress profile construction in the Yishu fault zone

### 4.1 Collection and collation of multisource stress data

The *in situ* stress data used in this paper are mainly obtained by hydraulic fracturing tests, which were conducted by our research group. Other stress data are from the published literature. The test system, test procedure, and data processing of the field hydraulic fracturing test were performed following the methods recommended by the International Society of Rock Mechanics (Haimson and Cornet, 2003). We summarized these data and evaluated the measured stress data in the study area

included in the latest version of the fundamental database of crustal stress environments in continental China (Xie et al., 2007). A total of 162 data pieces were obtained, the distribution of which is shown in Figure 5. The *in situ* stress measurement data were graded and screened according to the World Stress Map data quality classification system(Heidbach et al., 2016), and only test data with A, B and C quality based on the World Stress Map (WSM) ranking scheme and sounding depth >100 m were retained (Zoback, 2007).

To estimate the fault frictional strength and the in situ stress state in deep depth, we compiled earthquake focal mechanisms from the Institute of Geophysics, China Earthquake Administration, and the fundamental database of crustal stress environment in continental China. A total of 913 focal mechanism solutions were obtained in the study area and its surrounding areas for this study (Figure 1), with magnitudes ranging from Ms2.0 to Ms4.5, that is, focal mechanism solutions of small earthquakes, which were obtained by the first motion of *P* waves. A total of 117 focal mechanism solutions were obtained for the area near

the Yishu fault zone (longitude 117.55°–119.55°, latitude 35.00°–37.99°). These events occur throughout the seismogenic



crust up to a depth of ∼24 km, with an average focal depth of 9.4 ± 4.8 km. Among these, strike-slip events predominate, comprising 50% of the total compiled earthquakes, followed by reverse (23%) and normal (16%) faulting earthquakes.

Based on the measured stress data, the $\mu_m$ value of a fault can be obtained according to Eq. 6. Thus, the friction strength $\mu$ of shallow crustal faults can also be estimated. The calculation results based on the stress data (>100 m) measured by three

boreholes for three consecutive years in the study area show that the mean value of $\mu_m$ in the shallow crust fluctuates between 0.28 and 0.36 with a standard deviation between 0.06 and 0.12 (Table 2). The results of $\mu_m$ calculated at different depths are shown in Figure 5.

**Table 2: The measured $\mu_m$ value of three boreholes for three consecutive years**

| No. | Depth /m | $\mu_m$ | | | Mean $\mu_m$ |
| --- | --- | --- | --- | --- | --- |
| | | 2012 | 2013 | 2014 | |
| SLZK | 104.3−111.1 | 0.38±0.05 | 0.25±0.08 | 0.38±0.04 | 0.34±0.06 |
| HHZK | 178.1−395.5 | 0.29±0.08 | 0.28±0.09 | 0.28±0.09 | 0.28±0.09 |
| QSZK | 106.7−354.4 | 0.38±0.12 | 0.34±0.11 | 0.38±0.12 | 0.36±0.12 |

The value of $\mu$ in deep depth can be estimated from the angle $\theta$ between the $P$ axis of the focal mechanism solution and the fault plane (Iio, 1997). Since more than 700 values of $\mu$ are calculated by the focal mechanism solution (after the removal of unreasonable values), the $\mu$ values are averaged at depth intervals of 1 km, as shown in Figure 5a. The mean value of $\mu$ calculated from the angle $\theta$ between the $P$ axis of the focal mechanism solution and the fault plane is 0.37 ±0.23. This result is also consistent with the results of the macroscopic analysis (Figure 6), where a narrow peek in the $P$-axis distribution indicates

a smaller value of $\mu_m$. Therefore, the friction strength of the deep fault can also be approximated as 0.37 ±0.23.

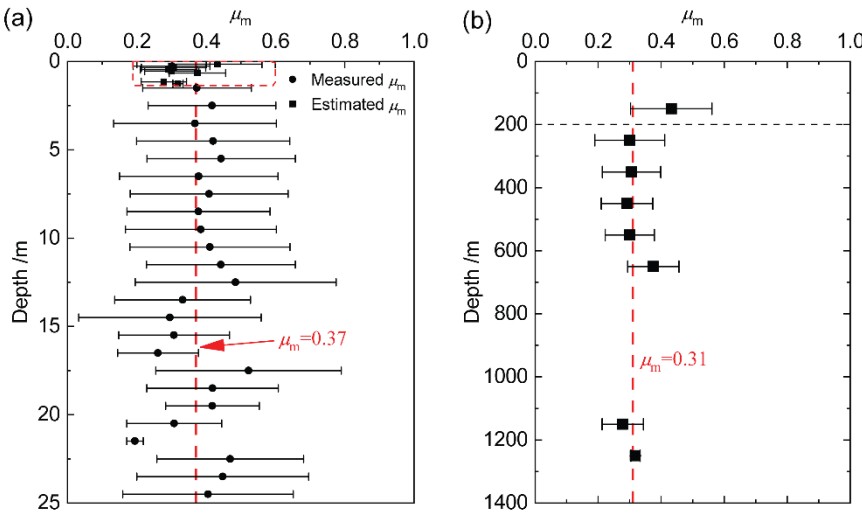




**Figure 5: The estimated value of $\mu_m$ in the Yishu fault zone. (a) $\mu_m$ estimated from focal mechanism solutions and measured data. (b) $\mu_m$ value based on measured in situ stress data at shallow depth.**

As shown from the calculation results in Figure 5, the value of $\mu_m$ at shallow depth calculated from the measured stress decreases with increasing depth. The mean value of $\mu_m$ becomes stable at approximately 200 m, and the mean value after stabilization is $0.31 \pm 0.09$. The mean value of $\mu_m$ in deep crust calculated from an angle $\theta$ between the direction of the $P$ axis and the fault plane is relatively close to $\mu_m$ at shallow depths. Therefore, the friction strength value of the fault at deep depth in the Yishu fault zone is approximately 0.37.

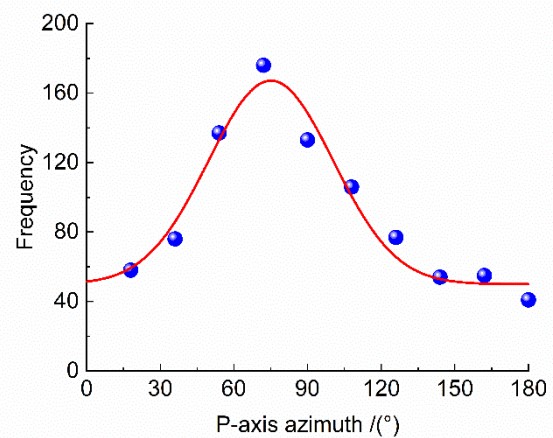


**Figure 6: The azimuth distribution of the $P$-axis. The distribution of the $P$ axes shows a narrow peek that indicates a smaller value of $\mu$**

### 4.2 Characteristics of tectonic stress in the deep and shallow parts of the Yishu fault zone

For in situ stress in shallow depth, Sheorey (1994) proposed an elastostatic thermal stress model to estimate crustal stress. The

average horizontal-to-vertical stress ratio can be given by(Sheorey, 1994):

$$K = 0.25 + 7E(0.001 + 1/z) \tag{8}$$

where $E$ refers to the average elastic modulus of the rock at a certain depth and $z$ is the burial depth in meters. Therefore, we use this equation to fit the measured data in the study area and estimate the stress magnitude at shallow depth.

We attempt to constrain in situ stress magnitudes in deep depth using multisource data and stress state limitation by Coulomb

frictional sliding on optimal-orientated faults (Zoback, 2007). Vertical stress is calculated from the overburden. The stress polygon allows us to estimate the ranges of possible magnitudes of maximum and minimum horizontal principal stresses at a particular depth for a given pore pressure and assumed coefficient of friction (Zoback, 2007). We assume that the stress state at each depth is characterized by consistent $K_H = S_H/S_V$ and $K_h = S_h/S_V$ ratios (horizontal-to-vertical stress ratios). Then the stress polygon could be illustrated by the horizontal stress ratio polygon (Wang et al., 2019; Wang et al., 2021). However, the

stress magnitudes estimated by stress polygon are usually in a wide range. To further constrain the stress magnitude in deep depth, Wang et al. (2019) incorporated the stress shape ratio $R$ $(=(S_1 - S_2)/(S_1 - S_3))$ inversed from focal mechanisms in stress



estimation by the stress polygon. Based on Anderson's fault theory, the relation between the stress shape ratio $R$ and the horizontal stress ratio could be expressed as (Wang et al., 2019):

$$\begin{cases} k_{\mathrm{H}} = Rk_{\mathrm{h}} + 1 - R & \text{(NF)} \\ k_{\mathrm{H}} = \dfrac{1}{1-R}k_{\mathrm{h}} + \dfrac{R}{R-1} & \text{(RF)} \\ k_{\mathrm{H}} = \dfrac{R}{R-1}k_{\mathrm{h}} + \dfrac{1}{1-R} & \text{(SS)} \end{cases} \tag{9}$$

where $k_{\mathrm{H}} = (S_{\mathrm{H}} - P_0) / (S_{\mathrm{V}} - P_0)$ and $k_{\mathrm{h}} = (S_{\mathrm{h}} - P_0) / (S_{\mathrm{V}} - P_0)$ are effective horizontal-to-vertical stress ratios. Therefore, we could further constrain the magnitude of the stress if we knew the range of the stress shape ratio.

The stress shape ratio $R$ can be estimated by the stress inversion of focal mechanisms (Vavryčuk, 2014). Based on the estimated stress shape ratio $R$, the horizontal-to-vertical stress ratios can be calculated using Eq. 9.

Based on the regional characteristics of the focal mechanism solutions of small earthquakes, we divided the study area into

three regions to construct stress profiles for subsequent fault stability analysis. For study area I (longitude 117 °–119 °, latitude 34 °–36 °), a simple fitting analysis was performed using the horizontal-to-vertical stress ratios $K$ (Figure 6(a)). As shown in Figure 7(a), the horizontal-to-vertical stress ratios follow Shoerey model in Eq. 8. The ranges of stress shape ratio in the study area and the adjacent areas are estimated by stress inversion of focal mechanisms (Table 3). Then the maximum and minimum horizontal-to-vertical stress ratios in study area I are calculated as $K_{\mathrm{H}} = 1.25 \pm 0.25$ and $K_{\mathrm{h}} = 0.81 \pm 0.19$, respectively. Combining

the results of the shallow and deep stress indicators in the area, a complete stress profile in the area was constructed (Figure 7(b)).

**Table 3: The estimated stress ratio for deep depth in the study area and adjacent area based on focal mechanisms**

| Place | Longitude | Latitude | $R$ | $K_{\mathrm{h}}$ | $K_{\mathrm{H}}$ |
|---|---|---|---|---|---|
| Study area I | 117°–119° | 34°–36° | 0.53±0.13 | 0.81±0.19 | 1.25±0.25 |
| Study area II | 117°–119° | 36°–39° | 0.76±0.12 | 0.87±0.13 | 1.40±0.40 |
| Study area III | 119°–121° | 36°–39° | 0.69±0.13 | 0.85±0.15 | 1.35±0.35 |
| Adjacent area 1 | 117°–119° | 32°–34° | 0.80±0.10 | 0.86±0.14 | 1.51±0.51 |
| Adjacent area 2 | 119°–121° | 32°–34° | 0.72±0.10 | 0.86±0.14 | 1.35±0.35 |
| Adjacent area 3 | 119°–121° | 34°–36° | 0.58±0.13 | 0.82±0.18 | 1.28±0.28 |
| Adjacent area 4 | 121°–123° | 32°–34° | 0.66±0.18 | 0.83±0.17 | 1.36±0.36 |
| Adjacent area 5 | 121°–123° | 34°–36° | 0.63±0.12 | 0.84±0.14 | 1.30±0.30 |
| Adjacent area 6 | 121°–123° | 32°–34° | 0.61±0.05 | 0.85±0.15 | 1.25±0.25 |



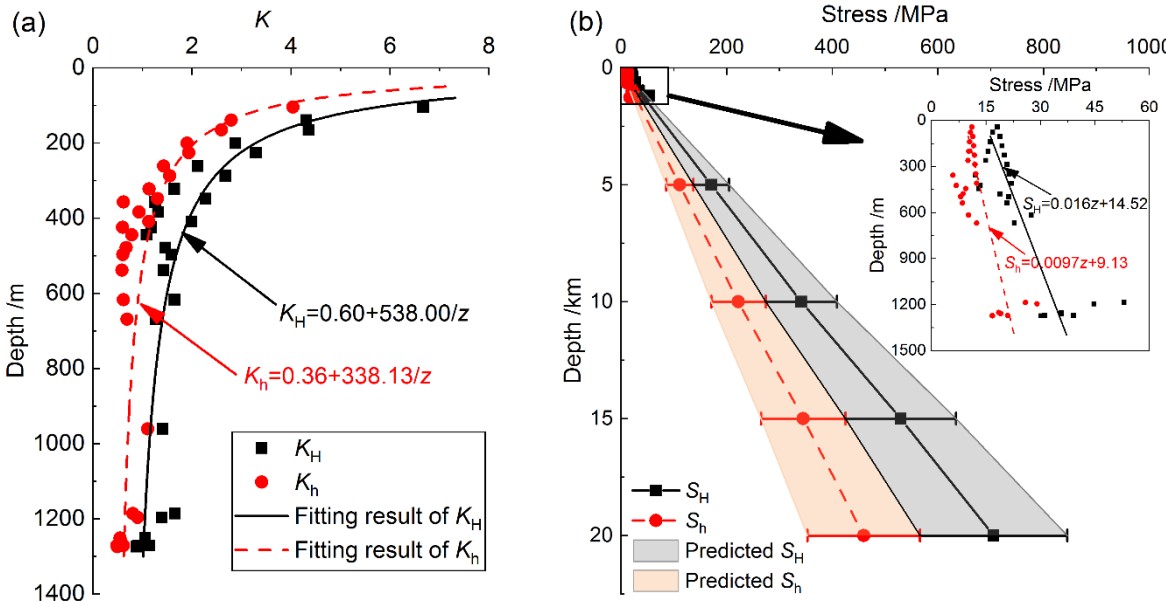

**Figure 7: Complete stress profile *versus* depth in study area I: (a) fitted horizontal-to-vertical stress ratios *versus* depth using Shoerey model, and (b) complete stress profile versus depth. $K_H = S_H/S_v$ and $K_h = S_h/S_v$ are the maximum and minimum horizontal-to-vertical stress ratios, respectively. The gray and pink shaded areas are the predicted possible ranges of the maximum and minimum horizontal stresses, respectively.**

As shown in Figure 7(a), the horizontal-to-vertical stress ratios $K_H$ and $K_h$ of study area I gradually decrease with depth, and $K_h$ gradually becomes less than 1 with increasing depth; that is, the stress regime changes from $S_v < S_h < S_H$ to $S_h < S_v < S_H$ with depth. The relationship between the magnitudes of the three principal stresses is stable below 600 m. Based on this trend, it can also be inferred that the deep stress regime favors strike-slip faulting. Horizontal-to-vertical stress ratios at shallow depths and the prediction results of the stress ratios at depth can be combined to produce a complete stress profile, as shown in Figure 7(b). The maximum (minimum) horizontal stresses predicted at 5 km, 10 km, 15 km, and 20 km are 170.76 ±33.76 MPa (111.23 ±85.47 MPa), 341.51 ±67.51 MPa (222.46 ±51.51 MPa), 529.85 ±104.75 MPa (345.13 ±265.21 MPa), and 705.46 ±139.46 MPa (459.54 ±106.41 MPa), respectively. The relative deviation of the maximum and minimum horizontal principal stresses in the prediction results are approximately 19.76% and 21.15%, respectively.

For study area II (longitude 117 °–119 °, latitude 36 °–39 °), the fitted results of the horizontal-to-vertical stress ratios are shown in Figure 8(a), and a complete stress profile in the area is shown in Figure 8(b). Figure 8(a) indicates that the horizontal-to-vertical stress ratios in study area II also show that $K_H$ and $K_h$ gradually decrease with depth and follow Sheorey model, with $K_h$ gradually becoming smaller than 1 with depth. The stress regime transition from $S_v < S_h < S_H$ to $S_h < S_v < S_H$ occurs with depth. The relationship between the magnitudes of the three principal stresses is stable below 200 m. This changing trend allows us to infer that the stress regime is in favor of strike-slip faulting in deep depth.

Analyses of the horizontal-to-vertical stress ratios at shallow depths and the prediction results of the horizontal-to-vertical stress ratios at deep depths, combined with the crustal velocity model crust1.0, suggest three principal stresses at different



depths and enable the construction of a complete stress profile as shown in Figure 8(b). Maximum (minimum) horizontal stresses estimated at 5 km, 10 km, 15 km, and 20 km are 165.83 ±47.32 MPa (102.71 ±15.79 MPa), 331.65 ±94.66 MPa (205.42 ±31.58 MPa), 792.05 ±226.06 MPa (356.24 ±54.76 MPa), and 705.46 ±139.46 MPa (490.58 ±75.41 MPa), respectively. The relative deviation of the maximum and minimum horizontal principal stresses in the prediction results are

approximately 28.54% and 15.37%, respectively.

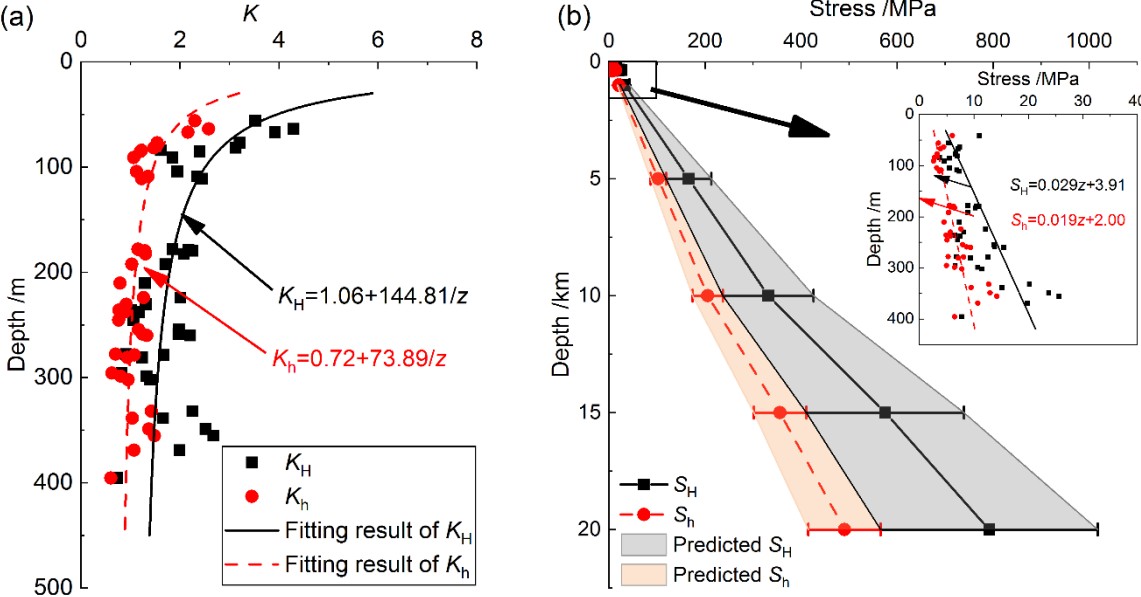

**Figure 8: Complete stress profile *versus* depth in study area II: (a) plot of calculated and fitted pressure coefficients *versus* depth, and (b) complete stress profile *versus* depth. $K_H$ =$S_H$/$S_V$ and $K_h$ =$S_h$/$S_V$ are the maximum and minimum horizontal-to-vertical stress ratios, respectively. The gray and pink shaded areas are the predicted possible ranges of the maximum and minimum horizontal**
**stresses, respectively.**

For study area III (longitude 119 °–121 °, latitude 36 °–39 °), the fitted results of horizontal-to-vertical stress ratios are shown in Figure 8(a). Combined with the results of the stress indicators in deep depth in the area, a complete stress profile in the area can be constructed, as shown in Figure 9(b).

As shown in Figure 9(a), horizontal-to-vertical stress ratios gradually decrease with depth, and $K_h$ gradually decreases to
become smaller than 1 with increasing depth. The $S_v < S_h < S_H$ stress regime converts to $S_h < S_v < S_H$ with depth. The relationship between the magnitudes of the three principal stresses is stable below 300 m, and it can also be inferred that the deep stress regime favors strike-slip faulting. Three principal stresses at different depths can be calculated, and a complete stress profile can be constructed, as shown in Figure 9(b). The maximum (minimum) horizontal stresses predicted at 5 km, 10 km, 15 km, and 20 km are 185.56 ±48.56 MPa (116.08 ±95.16 MPa), 371.13 ±97.12 MPa (232.16 ±41.84 MPa), 574.97 ±150.47 MPa
(359.68 ±64.82 MPa), and 766.63 ±200.63 MPa (479.57 ±86.43 MPa), respectively. The relative deviation of the maximum and minimum horizontal principal stresses in the prediction results are approximately 26.17% and 18.02%, respectively.





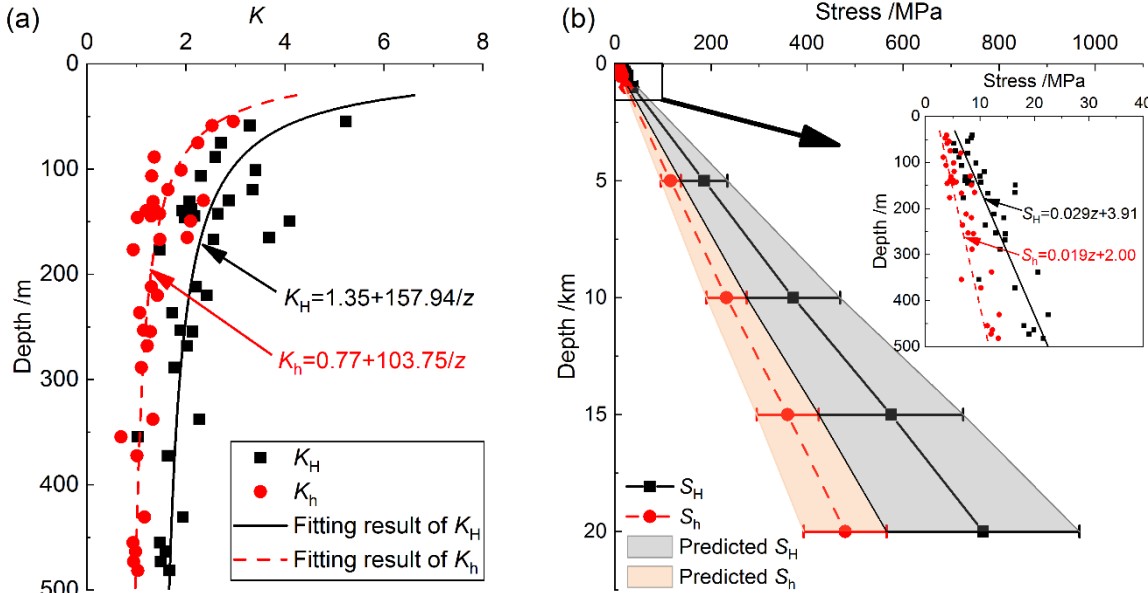

**Figure 9: Complete stress profile *versus* depth in study area III: (a) plot of calculated and fitted pressure coefficients *versus* depth, and (b) complete stress profile *versus* depth. $K_H = S_H/S_V$ and $K_h = S_h/S_V$ are the maximum and minimum horizontal-to-vertical stress ratios, respectively. The gray and pink shaded areas are the predicted possible ranges of the maximum and minimum horizontal stresses, respectively.**

## 5 Analysis of fault slip susceptibility for the Yishu fault zone

### 5.1 Analysis of fault slip susceptibility based on Mohr-Coulomb theory

According to the theoretical analysis presented in Section 2, the slip tendency of the fault is determined by the ratio of shear stress to normal stress (apparent friction strength) on the fault plane. Based on existing shallow stress data and the spatial distribution characteristics of each fault in the study area, Eq. 5 is used to calculate the apparent friction strength of each fault, allowing the slip susceptibility of the fault to be quantitatively analyzed using Byerlee's theory. The calculation of apparent friction strength and the analysis of Mohr circles were carried out in the three study areas. Vertical stress was used to perform the nondimensional comparative analysis of shear and normal stress.

When analyzing fault slip tendency under the action of the *in situ* stress field, considering that no artesian well or high-pressure aquifer paleobed is found in the study area, the pore pressure can be assumed to be equal to the hydrostatic pressure. Based on the results of the apparent friction coefficient on the fault plane, as shown in Figure 10, the apparent friction coefficient in study area I is mostly distributed between 0.13 and 0.30. The friction coefficient represented by the Mohr circle of mean stress (red) is 0.22, which is smaller than the rock friction coefficient given by Byerlee's theory (0.6–1.0), indicating that the stress state in study area I has not yet reached the critical stress threshold of faulting (according to Byerlee's theory). This indicates that the Yishu fault zone in this region is currently in a stable state. The fault activities of the other two study areas were analyzed similarly.



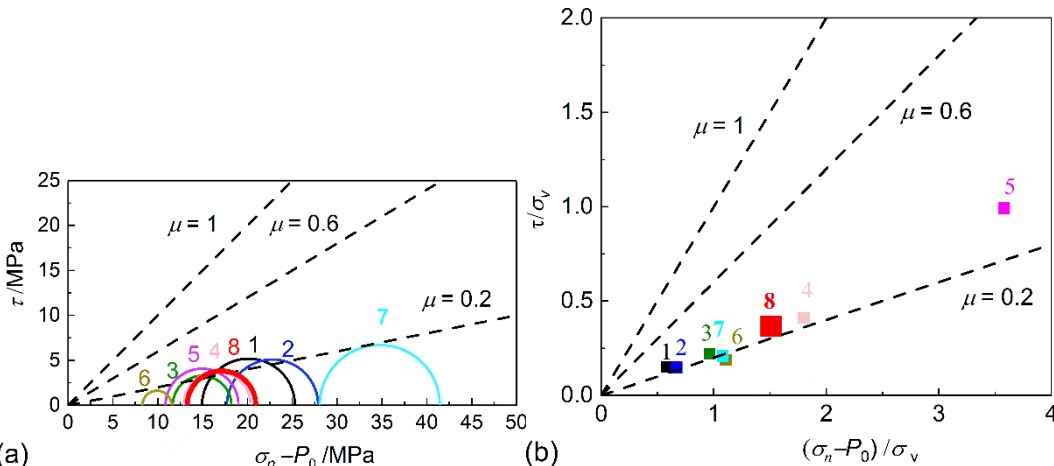

**Figure 10: Analysis of fault slip** susceptibility **under regional *in situ* stress for study area I. (a) Mohr circles of effective fault stress.** 350 **(b) Friction coefficients. 1, Huafeng mine, Tai'an; 2, Suncun mine, Xintai; 3, Jibei mine; 4, Yangying mine, Liangshan; 5, Feixian, Linyi; 6, Feixian, Linyi; 7, Xingcun mine, Qufu; 8, average results for study area I.**

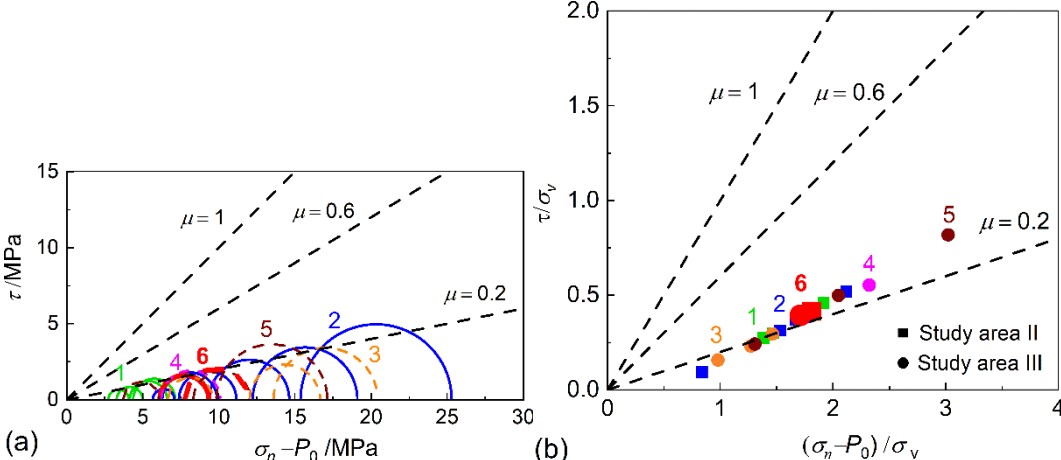

**Figure 11: Analysis of fault slip susceptibility for the stress field of study areas II and III. (a) Mohr circle of fault effective stress. (b) Friction coefficients. 1, SLZK borehole in Shanglin, Linju; 2, HHZK borehole in Honghe, Changle; 3, Zikou village, Cuizhao Road,** 355 **Shandong province; 4, Lingou, Penglai; 5, QSZK borehole in Qingshan, Changyi; 6, Regional average results for study areas II and III.**

Hydrostatic pressure was considered in the analysis of the fault slip under the *in situ* stress field acting in study areas II and III. The apparent friction coefficients in study area II are between 0.11 and 0.31, and the average value of friction strength is 0.20 (solid red line in the Mohr circle in Figure 10(a)). The calculated results of the apparent friction coefficients are smaller 360 than the rock friction coefficients given by Byerlee's theory; thus, it can be inferred that the faults in study area II are in a completely stable state. The apparent friction coefficients in study area III are mainly distributed between 0.16 and 0.29, with a mean value of 0.21. The red dotted line in Figure 10(a) represents the mean stress Mohr circle. All the calculated results of the apparent friction coefficients are also smaller than the rock friction coefficient given by Byerlee's theory. Therefore, the faults in study area III are also in a completely stable state.





## 5.2 Stress accumulation degree in the Yishu fault zone

The definition of the frictional characteristic index $\mu_m$ was introduced in detail above. The value of $\mu_m$ not only represents the degree of stress accumulation, but also roughly estimates the friction strength of a fault when the occurrence of the fault is uncertain. In this study, the shallow stress accumulation index obtained from the measured stress and the deep stress accumulation index estimated from the focal mechanism solutions are estimated, as shown in Figure 5. The distributions of $\mu_m$ in shallow and deep depth are generated and interpolated from scattered data, as shown in Figure 12.

Compared with the $\mu_m$ values in the deep and shallow depths, the degrees of stress accumulation in the shallow and deep crust of the Yishu fault belt generally have a similar distribution. The value of $\mu_m$ is medium to low in the main part of the Yishu fault belt, indicating that the degree of stress accumulation is low to moderate and the slip susceptibility of fault is low. The region near the northwest of the Yishu fault zone has a relatively high $\mu_m$ of 0.5, which represents a high degree of stress accumulation and is susceptible to fault slip. In contrast, the $\mu_m$ value in the southeast of the Yishu fault belt is less than 0.4, revealing a lower slip susceptibility of the fault.

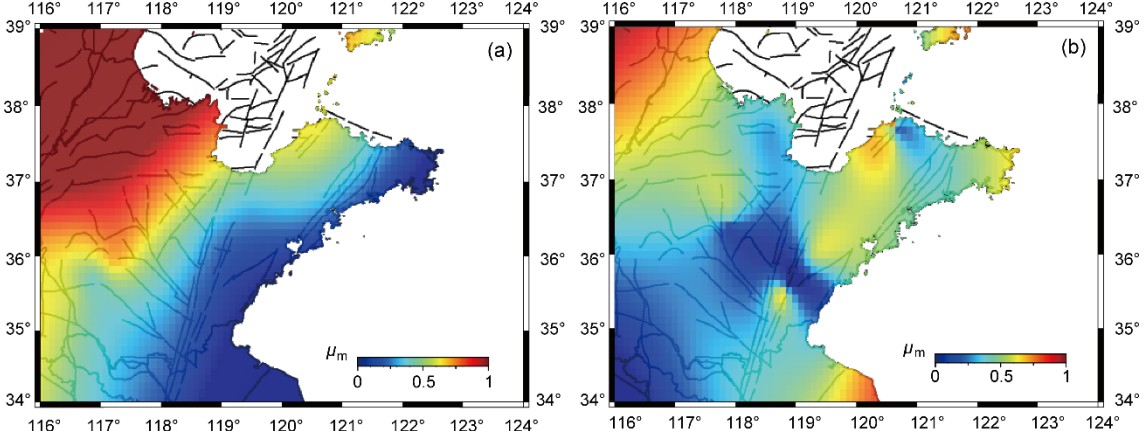

**Figure 12: Distribution of the frictional characteristic index $\mu_m$ distribution at two depths: (a) shallow and (b) deep crust.**

## 5.3 Comprehensive comparative analysis and evaluation

In the above analysis, we adopted the Mohr-Coulomb theory to compare and analyze the slip susceptibility of faults in three study areas. It is preliminarily concluded that the slip susceptibility of the faults in these portions of the Yishu fault zone is not higher than in the surrounding areas. When using Amontons' law and Coulomb's criterion as a basis for evaluating fault slip, it is found that the friction strength of the fault surface is the determining factor that directly affects slip activity.

In the three study areas, the stresses at different depths are constrained based on stress polygons and focal mechanism solutions. Consequently, the apparent friction strength $\mu$ can be calculated using the formula in Section 2 and is shown in Figure 13.



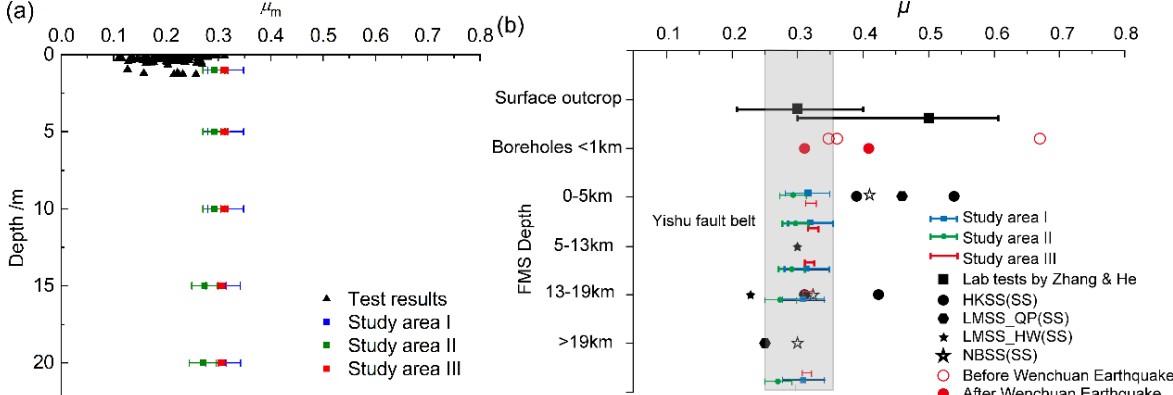

**Figure 13: Comparison between apparent friction strength *μ versus* depth in study areas and Longmenshan fault belts (Wang et al., 2015a; Zhang and He, 2013). (a) *μ* values of study areas *versus* depth calculated based on test results and predicted results. (b) Comparison of *μ* values in the Yishu and Longmenshan fault belts. The apparent friction strength *μ* in the study areas are mainly in the gray-shaded area.**

The friction strength ranges of the faults in study areas I, II and III (blue, green and red, respectively) at different depths are shown in Figure 13. It should be noted that since the obtained stress profile is a range value, the corresponding calculated friction strength is also an interval rather than a fixed value. The apparent friction strength $\mu$ in study areas I and II is in the range of 0.25~0.36, while it is in a narrow range of 0.30~0.32 in study area III. Figure 13(a) summarizes the apparent friction strength $\mu$ estimated based on measured data from the shallow portion of the Yishu fault zone, which are in the range of 0.11 to 0.31. The measured stress of the shallow part is readily influenced by surface denudation and underground engineering, among other factors; In fact, the predicted stress data at greater depths are more likely to reflect fault slip susceptibility at the focal depth. The range of friction strength of deep faults is consistent with those of shallow faults, which indicates that deep faults may be less susceptible or active.

To further illustrate the fault susceptibility to slip in the Yishu fault zone, we compared the slip susceptibility of fault in the Yishu fault zone with that in the Wenchuan earthquake area, Longmenshan fault belt. We compiled the results of apparent friction strength at different depths obtained by laboratory tests (Zhang and He, 2013), borehole observation (Wang et al., 2015a), focal mechanism analysis, and other methods in the Wenchuan Earthquake zone, Longmenshan fault belt (Figure13(b)). These results generally include not only the period prior to the main earthquake but also some results obtained at the time of the earthquake. To improve comparability, only the results of strike-slip faulting were selected from the collected results. The deep fault friction strength of the Yishu fault zone can be found to be distributed between 0.25 and 0.36, which is less than the fault friction strength before the Wenchuan earthquake (Wang et al., 2015a). This also shows that the apparent friction strength of the deep fault of the Yishu fault zone is generally lower than suggested by previous research results. Therefore, it can be concluded that the overall slip susceptibility of the fault in the Yishu fault zone is low. The slip susceptibility of the fault in the study area is relatively high in study area I and relatively low in study areas II and III, which is generally consistent with the results of previous analyzes.



## 6 Discussion

The value of $\mu_m$ can represent the degree of stress accumulation as presented in Section 5.2. To further illustrate the degree of stress accumulation and fault slip susceptibility, two typical case studies are presented from the point of view of shallow stress

accumulation, namely the Wenchuan earthquake and the Kobe (Hyogo-ken Nanbu) earthquake, Japan, as shown in Figure 14. Tanaka et al. (1998) studied the $\mu_m$ for approximately 20 years before the Kobe (Hyogo-ken Nanbu) earthquake, which occurred with magnitude 7.2 on January 17, 1995. Their results demonstrated that $\mu_m$ had been increasing before the earthquake and were close to $\mu_m = 0.6$ immediately before the event at both the Hiraki and Hoden sites (Figure 14(a)). However, measurements showed that it decreased dramatically after the earthquake at these two sites to the previous level of $\mu_m = 0.2$.

Wang et al. (2015) also studied $\mu_m$ in the months before and after the Wenchuan earthquake on May 12, 2008. The value of $\mu_m$ was 0.43 before the Wenchuan earthquake. However, it decreased significantly to 0.31~0.36 after the earthquake. Comparing the value of $\mu_m$ before the earthquake in these two regions shows that the accumulation of stress on the fault plane in the three study areas is much lower than that required for the occurrence of large earthquakes, indicating that the Yishu fault zone is not susceptible to slip.

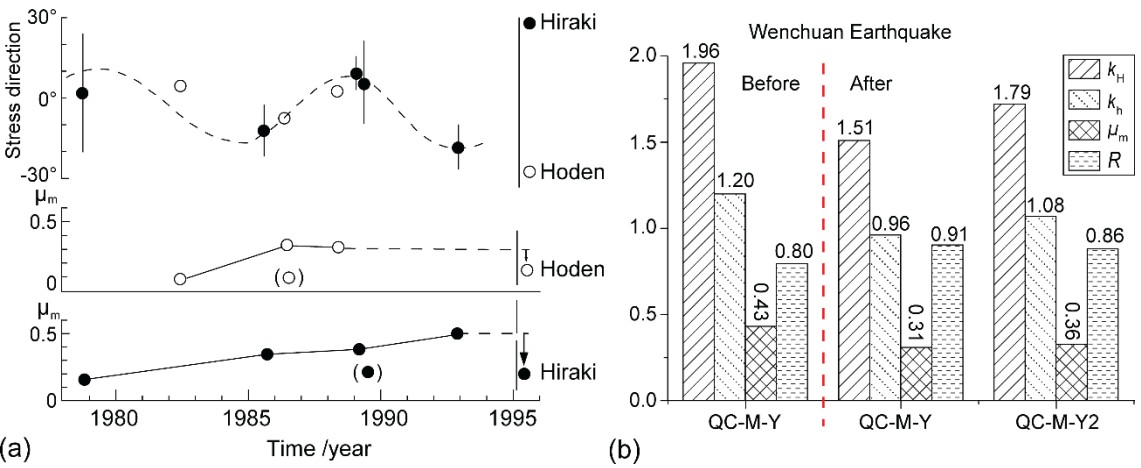

**Figure 14: Stress indicators change before and after typical earthquakes (Tanaka et al., 1998; Wang et al., 2015a). (a) Changes of**
**$\mu_m$ and variation of the direction of principal stress at the Hiraki (●) and Hoden (o) sites before and after the Kobe (Hyogo-ken**
**Nanbu) earthquake, Japan. (b) Changes in horizontal-to-vertical stress ratios ($k_H$ and $k_h$), $\mu_m$, and stress shape ratio $R$ before and**
**after the Wenchuan Earthquake in the QC-M-Y and QC-M-Y2 boreholes. The two measurement sites were located in the**
**mountainous area of Muyu village, Qingchuan County, 7 km from the seismogenic fault.**

To better analyze slip susceptibility or seismic risk in the Yishu fault zone, we incorporate other factors influencing seismic risk, including stress indicators, such as the horizontal-to-vertical stress ratio $K$ and angle $\theta$ between the $P$ axis and the fault plane, as well as the S-wave velocity and small earthquake density (Xu et al., 2017). The weight of each influencing factor may be determined using the analytic hierarchy process (AHP) and the expert scoring method (Vaidya and Kumar, 2006), as

listed in Table 4. After obtaining the weight of each influencing factor, it is necessary to perform information processing on the data of each influencing factor. Among these factors, the degree of stress accumulation $\mu_m$ has been estimated in detail and visualized in the previous section. Other influencing factors are processed and visualized as shown in Figure 15.





**Table 4: Weight of each influencing factor for slip susceptibility in the Yishu fault zone.**

| Influencing factor | $\mu_m$ | $K$ | S-wave velocity | $\theta$ | Small earthquake density |
|---|---|---|---|---|---|
| Weight | 0.219 | 0.177 | 0.122 | 0.133 | 0.347 |

**Figure 15: Distributions of the influencing factors of seismic risk in the study area: (a) $\mu_m$ in shallow depth, (b) horizontal-to-vertical stress ratio $K$, (c) S-wave velocity in 60 km depth from crust1.0 and Zhang (2005), (d) angle $\theta$ between the P axis and the fault plane calculated from FMS, and (e) small earthquake density.**



After determining the weight of each influencing factor, it is necessary to provide a unified score for each. A higher score indicates a factor more conducive to fault activity. Thus, the corresponding evaluation model is established according to the weight value and score of the influencing factor as follows:

$$SLIP = \sum_{i=1}^{n} P_i \times A_i \tag{10}$$

where $SLIP$ denotes the coefficient of activity of a geological structure, $P_i$ is the weight of each influencing factor, $A_i$ is the normalized result of each influencing factor and $n$ represents the total number of influencing factors. The overall estimated fault activity coefficient is presented in Figure 16. According to the classification criteria, the slip susceptibility of a fault is low when the $SLIP$ activity index is between 0 and 0.3, the slip susceptibility is medium when the $SLIP$ activity index is between 0.3 and 0.6, and a fault is highly susceptible to slip when the $SLIP$ activity index is greater than 0.6. The detailed

results of the slip susceptibility of geological structure and activity zoning are presented in Figure 16.

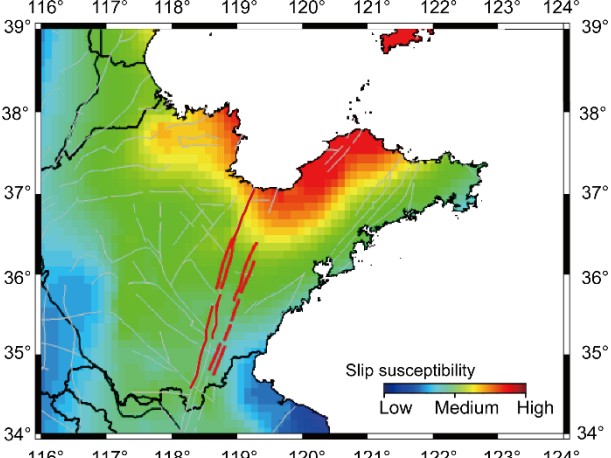

**Figure 16: Results of the comprehensive superposition analysis of the slip susceptibility of the fault in the Yishu fault zone.**

The results of the comprehensive superposition analysis of the five influencing factors are shown in Figure 16. These results indicate that, under the effect of the current regional stress state, the Yishu fault zone (red) faces risks associated with fault

activity. Different areas near the fault zone are characterized by variable fault slip susceptibility, with high slip susceptibility identified mainly in the northeast portion of the Yishu fault zone. The northern end of the fault zone is also an area of transition between high and medium risk. The central part of the Yishu fault zone and regions with moderate slip susceptibility generally remain stable, whereas the southeast and southwest parts of the Yishu fault zone have low slip susceptibility and low seismic risk. Currently, the Anqiu-Juxian fault is recognized as the latest active fault (Chao et al., 1998). In the regions considered by

this study, only the Anqiu-Juxian fault and the northern extension of two faults in the west graben are in, or close to, high susceptibility and high risk areas. The Yishui-Tangtou and Tangwu-Gegou faults of the west graben are all at moderate risk of fault slip. The Changyi-Dadain fault at the east boundary of the east graben is located in the transition zone between low- and





medium-risk areas. Therefore, our results are consistent with the currently accepted designation of active faults in the study area and are significant for regional crustal dynamics.

**7 Conclusions**

Using multi-source stress data in the Yishu fault zone, we constructed stress profiles and predicted the crustal dynamics of the Yishu fault zone based on slip susceptibility analysis. Our main conclusions are as follows:

(1) Based on multisource *in situ* stress data, the apparent friction of faults obtained using the Byerlee-Anderson method in the study area is mainly less than 0.6. For shallow depth, the average apparent friction coefficient of faults in study area I is 0.22,

while the average apparent friction coefficients of study areas II and III are 0.20 and 0.21, respectively. For deep depth, the apparent friction strength $\mu$ in study areas I and II is in the range of 0.25~0.36, while it is in a narrow range of 0.30~0.32 in study area III. According to the Byerlee-Anderson theory, the faults in all three regions are in a stable state; that is, fault slip is relatively difficult.

(2) The results of $\mu_m$ show that the overall degree of stress accumulation in the deep and shallow crust of the Yishu fault zone

is relatively low. $\mu_m$ in the northeast near the fault zone is relatively high, whereas $\mu_m$ in the southeast is relatively low. This is consistent with the results of the horizontal-to-vertical stress ratio at greater depths and proves that the horizontal-to-vertical stress ratio in deep depth is more reliable than that at shallow depths. It can also be preliminarily concluded that the Yishu fault zone does not have a high slip susceptibility.

(3) A comprehensive analysis of slip susceptibility of the fault in the Yishu fault zone was performed based on stress indicators

such as $\mu_m$, $K$, and $\theta$, as well as other seismic influencing factors, including S-wave velocity and seismic density. The results based on these five influencing factors indicate that the northwestern Yishu fault zone has a high seismic risk and may be the main seismograph region in the future. The overall seismic risk in the central part of the Yishu fault zone is not high. The southeastern part of the Yishu fault zone is close to a low-value area of seismic risk associated with the transition zone from low to medium seismic risk and exhibits the lowest risk of fault slip throughout the entire fault zone studied.

**Acknowledgments**

This work was supported by the National Natural Science Foundation of China (42174118) and a research grant (ZDJ2020-07) from the National Institute of Natural Hazards, Ministry of Emergency Management of China.

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
