# Peer review of "Prediction of Crustal Dynamics for the Yishu Fault Zone Based on Slip Susceptibility Analysis"

_EGUsphere, 2023_

## Referee Comment (RC1)

This paper estimated the slip susceptibility of faults in the Yishu fault zone using a slip tendency analysis based on coupled tectonic stress profiles.The conclusions have certain scientific significance on studying the geomechanical and fault mechanics evidence for evaluating the regional crustal dynamics of the Yishu fault zone, in North China. The manuscript has merit but requires revison before it can be accepted for publication, as follows:

(1) In "1. Introduction and 4.1. Collection and collation of multisource stress data". There are a large number of earthquake focal mechanism solutions with different magnitudes and types (as shown in Fig.1) , the authors should elaborate the data source, the data quality and the quantity. The results of these focal mechanism solutions should be also provided as a supplement.

(2) In "3. Regional geological background and fault structure". It's suggested that some representative geophysical profiles about the structures of the Yishu fault belt should be added.

(3) In "4.1. Collection and collation of multisource stress data". As shown in Fig.5, the $\mu_m$ varies between 0.31 and 0.37. The results indicate that the friction strength of the Yishu fault is relatively lower than the empirical value of 0.6. The $\mu_m$ is estimated using in situ stress measurements at shallow depths. The friction strength of the fault can be obtained by the rate- and state-friction experiment on the fault gouges. What's the relationship between the $\mu_m$ and the friction coefficient from the experiments. As is known to all, frictional stability, as related to

earthquake nucleation, is typically presented by the rate- and state-friction equation. Besides, in Table.2, the locations of three boreholes are not clear in Fig.1.

(4) In "4.2.Characteristics of tectonic stress in the deep and shallow parts of the Yishu fault zone". Three areas of I, II and III should be marked in Fig.1,and the authors should give some considerations to these study areas, just classified by longitude and latitude, or others?. In Figures 7,8 and 9, how to obtain the the maximum and minimum horizontal stresses profile versus depth at the depth interval of 400~1500 m. If these results are based on previous research, please provide various citations or these in situ stress measurements. As shown in Equation 9, the stress shape ratio $R$ will be estimated by the stress inversion of focal mechanisms, then the relationship between the $K_H$ and $K_h$ can be obtained, subsequently. Using the Equation 8, the average horizontal-to-vertical stress ratio can also be estimated. By combining formula 8 and formula 9, the authors obtain the results of $K_H$ and $K_h$ near the Yishu fault belt. The stress shape ratio was estimated using focal mechanisms in study area I, II, and III (a small scale), however, the average horizontal-to-vertical stress ratio was calculated by Sheorey's empirical equation (a crustal stress scale). Under these assumptions, how to ensure the difference and reliability of calculation results near the fault belt?. It is suggested that the average horizontal-to-vertical stress ratio should be estimated using empirical equation in North China or in Shandong province.

(5) In "5.1 Analysis of fault slip susceptibility based on

Mohr-Coulomb theory". The slip tendency of the fault is determined by the ratio of shear stress to normal stress (apparent friction strength) on the fault plane using in situ stress measurements at different boreholes. The authors should elaborate the calculated depth, the target fault plane, the values of principal stresses, and the orientation of the maximum principal stress. Besides, the locations of these boreholes used in calculation should be also marked in Fig.1 or some other figures. As shown in Figures 10 and 11, the average value of friction strength is taken as 0.20, which is different from the 0.37 (line 240). Why not assess the the slip tendency of the fault using 0.37?. In addition, if the value of 0.20 has been taken as a critical friction strength, large numbers of Mohr's circles (as shown in Fig.10 and 11) surpass the range of threshold value, indicating an unstable state. This phenomenon could have skewed the authors' results "a completely stable state". The authors should discuss this significant uncertainty.

(6) In "5.2 Stress accumulation degree in the Yishu fault zone". The authors should illustrate how to calculate the frictional characteristic index, such as the depth, the value of principal stresses. As shown in Fig.1, some study areas are lack of focal mechanisms or in situ stress measurements, however, Figure 12 shows the distribution of the frictional characteristic index in the entire region. How to get the accurate calculation results nearby?.

(7) In "Discussion". It seems that the results of stress indicators change before and after typical earthquakes as shown in Fig.14 are

unrelated to the study of frictional stability of the Yishu fault. Whether the similar stress indicators changes have been observed near the Yishu fault belt?. The authors assess the comprehensive superposition analysis of the slip susceptibility of the fault activity. Some factors are taken into consideration. This paper mainly analyzed the influence of in situ stress field on the faulting stability, including the magnitude of the principal stresses, and the orientation of the maximum principal stress. The other influenced factors associated with the earthuakes or faulting activity are not completely clear. The authors should discuss the rationality of the evaluation factors as shown in Fig.15. Besides, the earthquake number or small earthquake density should be the validation results, rather than being one of the influenced factors.

(8) Fig.16 shows the comprehensive superposition analysis of the slip susceptibility of the fault in the Yishu fault zone. In fact, the results are evaluated in the way of patchy distribution. The differential distributions of the faulting activity are rarely reflected. Therefore, the scientific significance of the Fig.16 could be limited to some extent.